# Sulfated Chinese Yam Polysaccharides Alleviate LPS-Induced Acute Inflammation in Mice through Modulating Intestinal Microbiota

**DOI:** 10.3390/foods12091772

**Published:** 2023-04-25

**Authors:** Shihua Wu, Xianxiang Chen, Ruixin Cai, Xiaodie Chen, Jian Zhang, Jianhua Xie, Mingyue Shen

**Affiliations:** State Key Laboratory of Food Science and Technology, Nanchang University, Nanchang 330047, China; 18222591639@163.com (S.W.); xianxiangchen@email.ncu.edu.cn (X.C.); c18312933692@163.com (R.C.); xdiechen@163.com (X.C.); zj105716@163.com (J.Z.); jhxie@ncu.edu.cn (J.X.)

**Keywords:** yam polysaccharide, sulfation, LPS, anti-inflammation, gut microbiota

## Abstract

This study aimed to test the preventive anti-inflammatory properties of Chinese yam polysaccharides (CYP) and sulfated Chinese yam polysaccharides (SCYP) on LPS-induced systemic acute inflammation in mice and investigate their mechanisms of action. The results showed that SCYP can efficiently reduce plasma TNF-α and IL-6 levels, exhibiting an obvious anti-inflammation ability. Moreover, SCYP reduced hepatic TNF-α, IL-6, and IL-1β secretion more effectively than CYP, and significantly altered intestinal oxidative stress levels. In addition, a 16S rRNA gene sequencing analysis showed that CYP regulated the gut microbiota by decreasing *Desulfovibrio* and *Sutterella* and increasing *Prevotella*. SCYP changed the gut microbiota by decreasing *Desulfovibrio* and increasing *Coprococcus*, which reversed the microbiota dysbiosis caused by LPS. Linear discriminant analysis (LDA) effect size (LEfSe) revealed that treatment with CYP and SCYP can produce more biomarkers of the gut microbiome that can promote the proliferation of polysaccharide-degrading bacteria and facilitate the intestinal de-utilization of polysaccharides. These results suggest that SCYP can differentially regulate intestinal flora, and that they exhibit anti-inflammatory effects, thus providing a new reference to rationalize the exploitation of sulfated yam polysaccharides.

## 1. Introduction

Inflammation is a complicated physiological phenomenon which can be triggered by either infectious or non-infectious conditions [1,2]. Cytokines, such as tumor necrosis factor (TNF)-α and interleukin (IL)-6, have been known to play significant roles in inflammatory responses. They are released from cells, which causes the expression of adhesion molecules to recruit lymphocytes, monocytes, and neutrophils, and then move out of vessels to tissues, such as the liver, and induce inflammation and oxidative stress [3]. Lipopolysaccharide, an ideal inflammatory molding agent, derives from the cell wall of gram-negative bacteria and is an important component of the bacterial outer membrane, which can be recognized by the innate immune system [4]. Lipopolysaccharide entering the blood can trigger a systemic inflammatory response and may further cause multi-organ damage and dysfunction in the liver, lungs, intestines, kidneys, and brain, which can lead to clinically identified sepsis [5]. To alleviate inflammation, researchers have tried to find biologically active substances of natural origin. Many plant-derived natural compounds exhibit good biological activities with low toxic effects, and polysaccharides are one of the macromolecules that have received much attention for this reason [6].

Many plant-derived natural compounds exhibit good biological activities with low toxic effects, and polysaccharides are one of the macromolecules that have received much attention for this reason [6,7,8]. It has been demonstrated that polysaccharides perform a variety of biological activities including antioxidant, anti-inflammatory, antitumor, immunomodulatory, and antiviral activities [9,10], and they can be used in the preparation of new drugs, medicinal materials, and functional foods [9,11]. As a traditional medicinal food homologous plant, Chinese yam (*Dioscorea opposita* Thunb.) is rich in a variety of nutrients and bioactive components, such as polysaccharides, allantoin, saponins, and glycoproteins, which perform various biological activities and have various pharmacological effects [12]. The polysaccharides in yams are usually divided into starch and non-starch polysaccharides, and the yam polysaccharides used in this study were mucilage polysaccharides (non-starch polysaccharides) [13]. The different biological activities of polysaccharides are related to their different molecular weights, molecular structures, and three-dimensional structures, and these can be changed by structural modifications to alter their original active effects [6,14]. Sulfated polysaccharides are polysaccharides containing sulfate groups on the polysaccharide units. Some polysaccharides are naturally sulfated polysaccharides, and some polysaccharides do not originally have sulfate groups, though these can be prepared by chemical modification of sulfation sites [14]. Because of their own characteristics, sulfated polysaccharides perform many biological activities, such as anticoagulant, hypolipidemic, antiviral, antitumor, and antioxidant activities [15,16,17,18]. In recent years, synthetic sulfated polysaccharides have attracted attention for their structural modifications, by which their biological activities may be enhanced [19]. Aside from the Wolfrom method and the concentrated sulfuric acid and sulfur trioxide-pyridine method, some researchers prepare arabinogalactan sulfates with sulfur content up to 11.3% using ammonium sulfamate as a sulfating reagent [15]. In addition, few of the non-starch polysaccharides are digested by the mammalian intestine and reach the colon intact to serve as an energy source for the intestinal flora, stimulating their growth and producing healthy metabolites [20,21].

The gut microbiota is a diverse ecosystem which exists in symbiosis with the human body and plays relevant roles in human health, impacting the pathogenesis of many diseases [22]. Certain bacteria can increase intestinal permeability, and their structural components can enter the organism and trigger a cytokine cascade response to inflammation [23,24]. Other bacteria can exert effects on the regulatory cells of the immune system, suppressing inflammation or indirectly performing anti-inflammatory functions through their metabolites [24]. When the intestinal barrier is damaged and intestinal permeability increases, some substances that are not supposed to cross the intestinal wall, which is a component of the cell wall of gram-negative bacteria, escape from the intestinal lumen into the body’s circulation [22]. Bacterial translocation in the gut can cause damage to other tissues and organs, including the liver. The colonization of beneficial intestinal bacteria facilitates the alleviation of liver disease, and the health and homeostasis of the gut are important for liver health [22,24].

Our previous studies have demonstrated that sulfated derivatives of yam polysaccharides can regulate immune effects in RAW264.7 cells [25]. However, how sulfated derivatives of yam polysaccharides affect body inflammation and changes in the gut microbiota in mice is not well understood. In this study, the anti-inflammatory activities of CYP and SCYP were investigated by establishing an LPS-induced acute inflammation model in mice. The alleviating effects of both polysaccharides on the systemic inflammatory response, including blood circulation, in the liver and at intestinal sites, as well as the modulating effect on intestinal flora, were the main focuses. This research can provide a reference for the application of yam polysaccharides in food and medicine.

## 2. Materials and Methods

### 2.1. Materials

Lipopolysaccharide (LPS) derived from *E. coli* 055: B5 was purchased from Sigma-Aldrich (St. Louis, MO, USA). Acetylsalicylic acid (aspirin) was bought from Aladdin Bio-Chem Technology Co., Ltd. (Shanghai, China). Enzyme-linked immunosorbent assay (ELISA) kits were purchased from Boster Biological Technology Co., Ltd. (Wuhan, China). Total protein contents (BCA), superoxidase dismutase (SOD), catalase (CAT), and malondialdehyde (MDA) were purchased from Beyotime Biotechnology Co., Ltd. (Shanghai, China).

Yams were purchased from Ruichang (Jiangxi, China) and prepared as yam powder by crushing. The yam powder was depigmented, water was extracted, alcohol was precipitated, starch and protein were removed, it was dialyzed, alcohol was again precipitated, and it was redissolved and lyophilized to obtain CYP [19]. Referring to our previous study, SCYP was obtained by further sulfation modification of CYP using the chlorosulfonic acid pyridine method. The ratio of chlorosulfonic acid to pyridine was 1:5 and the degree of substitution (DS) was 0.44 [25]. The two polysaccharides were mainly composed of galacturonic acid, galactose, glucose, xylose, arabinose, and rhamnose. The CYP had a molar ratio of 2.77:1.41:0.98:0.91:0.27:0.18, and the SCYP had a molar ratio of 1.99:2.82:0.65:1.97:0.76:0.12 [25].

### 2.2. Animals and Experimental Design

Male C57BL/6J mice (7–8 weeks old, 22 ± 2 g body weight) were supplied by the Hunan Slac Jingda Laboratory Animal Co., Ltd. (Changsha, China, certificate number: SCXK (Xiang) 2016-0002). All the mice were kept at the animal laboratory of the State Key Laboratory of Food at Nanchang University. Before the experiments began, the mice underwent a 7-day adaptation period during which they were housed at a temperature of 18–22 °C, 55% relative humidity, and under a 12/12 h light–dark cycle with ad libitum access to a standard diet and water. The thirty mice were randomized into 5 treatment groups (6 mice per group): the normal saline contrast group (N), the LPS model group (M), the aspirin positive contrast group (P), the CYP group (CYP), and the SCYP group (SCYP). The specific treatment was as follows: the N and M groups were gavaged with an equivalent volume of normal saline, the P group was gavaged with aspirin (15 mg/kg BW), and the polysaccharide groups were gavaged with CYP and SCYP 100 mg/kg BW (body weight) once a day for 14 days. At the end of 14 days, the N group was injected with normal saline intraperitoneally, and the rest of the groups were injected with LPS (5 mg/kg BW) intraperitoneally for 12 h. After recording their weights, the mice were sacrificed by cervical dislocation.

All the animal experimental operations were carried out in accordance with the National Institutes of Health (NIH Publication No. 8023, 1978) guidelines for the care and use of laboratory animals, and all the experimental procedures were approved by Nanchang University Animal Ethic Review Committee (license No: SYXK (gan) 2015–0001).

### 2.3. General Indicators

After the mice were sacrificed, the thymus, spleen, and liver were removed and cleaned with 0.9% NaCl, and the excess solution was aspirated with filter paper and weighed, and the organ index was calculated. Organ index (mg/g) = organ weight (mg)/body weight (g).

### 2.4. Determination of Plasma Inflammatory Factors

Blood was collected from the eyes of the mice and centrifuged at 3000 rpm for 10 min at 4 °C. The supernatant was collected as plasma, aliquoted, and stored at −80 °C, and the assay was conducted according to the ELISA kit instructions.

### 2.5. Determination of Liver Inflammatory Factors

The liver was isolated from each mouse and cut into small pieces with scissors. Approximately 30 mg liver tissue was collected to make 10% (*m*/*v*) tissue homogenate with pre-cooled PBS, and the supernatant was collected by centrifugation at 12,000× *g* for 10 min at 4 °C and detected according to the ELISA kit instructions.

### 2.6. Oxidative Stress Factors and Jejunal Inflammatory Factors 

The small intestine was isolated from each mouse, the middle jejunal part was intercepted, and the contents were removed. Approximately 30 mg jejunal tissue was collected to make homogenate, and the centrifugal collection of supernatants was conducted in the same way outlined in Section 2.5. The protein concentrations in the collected supernatants were determined using a BCA kit, and SOD, CAT, MDA, and IL-1β levels were measured according to the kit instructions.

### 2.7. 16S rRNA High-Throughput Sequencing

After the mice were sacrificed, the colon of each mouse was isolated, the colonic contents were packed in EP tubes by extruding them with forceps, and these were stored in a −80 °C freezer. Total DNA was extracted from the colon contents according to the DNA extraction kit instructions. The extracted DNA was subjected to 0.8% agarose gel electrophoresis for molecular size determination and was quantified using a UV spectrophotometer. Bacterial 16S rRNA V3-V4 region-specific primers were used for PCR amplification. The primers were 338F: 5′-ACTCCTACGGGAGGCAGCA-3′ and 806R: 5′-GGACTACHVGGGTWTCTAAT-3′. The PCR products were purified using 2% gel electrophoresis, quantified using the Quant-iT PicoGreen dsDNA Assay Kit, and mixed according to the amount of data required for each sample. Library construction was performed using the Illumina’s TruSeq Nano DNA LT Library Prep Kit. The libraries were quality-checked on Agilent Bioanalyzer using the Agilent High Sensitivity DNA Kit and quantified on Promega QuantiFluor using the Quant-iT PicoGreen dsDNA Assay Kit. For qualified libraries with concentrations above 2 nM, 2 × 250 bp double-end sequencing was performed on a MiSeq machine using the MiSeq Reagent Kit V3 (600 cycles).

### 2.8. Statistical Analysis

Data were shown as the means ± standard deviation (SD). IBM SPSS statistical software 20 was used for statistical analysis. A data significance analysis was performed using one-way ANOVA and Duncan′s multiple range test. *p* < 0.05 or *p* < 0.01 was considered as significant.

## 3. Results

### 3.1. Body Weight and Organ Index

Throughout the entire experiment, changes in the body weight of the mice in each group were monitored (Figure 1A,B). After the intraperitoneal injection with LPS, a 9% weight loss was observed in the M group compared with the N group (*p* < 0.05). Compared with the M group, neither aspirin nor CYP treatment restored the weight loss caused by LPS. However, the SCYP group exhibited a significant (*p* < 0.05) improvement in body weight compared with the M group, indicating that the administration of SCYP could dramatically inhibit LPS-induced weight loss.

According to the results of the spleen index calculation, the spleens of the mice in group M were significantly enlarged by the intraperitoneal injection of LPS compared with those of the normal group (*p* < 0.05). This result is similar to that obtained in [26], in which it was found that the and spleen indices increased in the LPS-treated group compared with the untreated group. The spleen is the largest lymphatic organ and its functions include blood storage, the removal of aging red blood cells, and immune response, and it can therefore reflect the body’s inflammatory condition. Thus, the result revealed that the spleen was swollen, and the body was inflamed. Interestingly, this enlargement was not reduced by gavage with CYP, but with SCYP (*p* < 0.05). The liver index was significantly reduced in the M group compared with the normal group (*p* < 0.05). The CYP and SCYP were able to restore the hepatic index to a normal level compared with the model group (*p* < 0.05) (Table 1).

### 3.2. Plasma Inflammatory Cytokines

The results of the plasma pro-inflammatory cytokine changes in the mice are presented in Figure 1B,C. In normal mouse plasma, low levels of TNF-α and very low levels of IL-6 are typically detected. After LPS stimulation, TNF-α and IL-6 levels were significantly increased compared with those of the normal mice (*p* < 0.01). Compared with those of the M group, the TNF-α and IL-6 levels showed an extremely significant decrease in both the CYP-treated group and the SCYP-treated group (*p* < 0.01). The CYP group showed a stronger reduction plasma TNF-α and IL-6 levels than the SCYP group (*p* < 0.05). Overall, CYP and SCYP were effective in reducing both plasma TNF-α and IL-6 pro-inflammatory factors (Figure 1C,D).

### 3.3. Liver Inflammatory Cytokines

Compared with that of the N group, there was a significant increase in the level of TNF-α in the livers of the mice after LPS stimulation (*p* < 0.05) (Figure 2A). Compared with that of the M group, the CYP group showed a slight reduction in TNF-α level. However, the pre-administration of SCYP caused a significant reduction (*p* < 0.05). The IL-6 level was significantly increased in the model group compared with the N group (*p* < 0.05). Interestingly, pre-treatment with SCYP resulted in a significant recovery, and even a return to normal levels (Figure 2B). The level of IL-1β was significantly higher in the M group compared with the normal group (*p* < 0.05). CYP and SCYP had comparable effects; they both significantly reduced the level of IL-1β (*p* < 0.05) (Figure 2C). In general, CYP and SCYP both exhibited a great anti-inflammatory effect by decreasing the production of proinflammatory cytokines, and SCYP showed the better effect than CYP.

### 3.4. Jejunal Oxidative Stress Factors and Inflammatory Factors

Indicators related to oxidative stress in the jejunal segment of the small intestine were tested. For both of the antioxidant enzymes SOD and CAT, the model group showed a significant decrease compared with the N group (*p* < 0.05) (Figure 3A,B). The polysaccharide-treated groups had higher levels of both enzymes than the M group. For the lipid oxidation hazard MDA, the model group presented a significant elevation compared with the normal control group (*p* < 0.05) (Figure 3C). A slight reduction in MDA was found in the polysaccharide-treated group compared with the M group. A significant difference in the inflammatory factor IL-1β was not observed in any group (Figure 3D).

### 3.5. α-Diversity and β-Diversity Analysis of Colon Microbiota

Alpha diversity refers to the indicator of richness, diversity, and evenness of species, and larger Shannon–Simpson, Chao1, and observed indices represent higher within-habitat diversity. Four indices were used to evaluate the effect of LPS modeling and CYP and SCYP on the α-diversity of the intestinal microflora (Figure 4A). Compared with the N group, increases in the Shannon–Simpson, Chao1, and observed species indices (*p* < 0.05) of the intestinal microflora were observed in the CYP and SCYP groups. No significant difference in Simpson index was observed in any of the groups. This indicates that CYP could increase richness and evenness of the gut microorganisms.

A β-diversity analysis was used to evaluate the structural changes in the intestinal flora. A principal component analysis (PCA) based on Euclidean distance showed that the individuals in each experimental group exhibited significantly different differential clustering (Figure 4B). The N group was completely separated from the M, CYP, and SCYP groups on the *x*-axis (87.9%), indicating that LPS treatment greatly altered the intestinal structure of the mice. The CYP and SCYP groups were mostly separate from the M group and mostly overlapped with the N group on the *y*-axis (11.9%). This suggests that CYP and SCYP treatment affords partial resistance to LPS-induced alterations in gut microflora structure, and these groups increasingly resembled the normal group. This result was similar to the aforementioned results in which pre-treatment with polysaccharides decreased proinflammatory factors when the compared with the absence of such treatment. This might be because the intestinal flora was not seriously damaged by LPS.

### 3.6. Microbial Community Composition

According to the phylum-level species composition diagram (Figure 5A), the top four represented bacteria were *Bacteroidetes*, *Firmicutes*, *Proteobacteria*, and *TM7* (in that order). The specific relative abundance values of each phylum in each group are shown in Figure 5B–E. The relative abundance of *Bacteroidetes* was significantly increased in the M, CYP, and SCYP groups compared with the N group (*p* < 0.05). The relative abundance of *Proteobacteria* increased significantly in the LPS group compared with the N group (*p* < 0.05). However, the relative abundance was decreased by the polysaccharide treatment. A significant decrease in the relative abundance of *Firmicutes* and *TM7* was observed in the LPS group compared with the N group (*p* < 0.05). Interestingly, polysaccharide treatment can reverse the change in gut microbiota caused by LPS.

The overall species composition of the top 20 bacteria at the genus level is shown in Figure 6A, and the specific relative abundance values for each genus are shown in Figure 6B–G. The relative abundance of *Lactobacilllus* was significantly reduced in the LPS group compared with the N group (*p* < 0.05), and preventive administration of both polysaccharides did not restore the relative abundance. The relative abundances of *Shigella*, *Desulfovibrio*, and *Sutterella* were significantly elevated in the LPS group compared with the N group (*p* < 0.05). Compared with group M, shigella was less abundant in the CYP and SCYP groups, but the difference was not significant. *Desulfovibrio* was significantly reduced in both the CYP and SCYP groups, in which it reached the level of group N (*p* < 0.05). Moreover, the relative abundance of *Sutterella* was significantly reduced in the CYP group (*p* < 0.05). There was no significant difference in *Prevotella* in the N, M, and SCYP groups, though a significant elevation was observed in the CYP group compared with these (*p* < 0.05). There was no significant difference in the relative abundance of *Coprococcus* in the N, M, and CYP groups, though a significant elevation was observed in the SCYP group compared with these (*p* < 0.05).

### 3.7. Biomarker Bacteria

A linear discriminant analysis effect size (LEfSe) combining the rank sum test and a discriminant analysis was used to identify significantly changed microbial taxa. The cladogram demonstrates the taxonomic hierarchical distribution of marker species in each group of samples, with larger solid nodes representing more significant enrichment (Figure 7A). Setting 4 as the LDA score threshold, the histogram shows the biomarker bacteria and their significance, with longer bar lengths representing more significant differences (Figure 7B). In group N, p_*Firmicutes*, f_*Lactobacillaceae*, o_*Lactobacillales*, c_*Bacilli*, g_*Lactobacillus*, and f_*Rikenellaceae* were specific. p_*Proteobacteria*, c_*Gammaproteobacteria*, f_*Enterobacterianceae*, o_*Enterobacteriales*, and g_*Shigella* were the biomarkers in group M. The CYP group was enriched with p_*Bacteroidetes*, c_*Bacteroidia*, o_*Bacteroidales*, f_*Prevotellaceae*, g_*Prevotella*, f_*Flavobacteriaceas*, o_*Flavobacteriales*, and c_*Flavobacteriia*. The SCYP group was enriched with f_*Bacteroidaceae*, g_*Bacteroides*, and g_*Pasteurella*.

## 4. Discussion

Inflammation is part of a protective reaction against damage caused by invading microorganisms. It is essential to trim or adjust it to avoid any increasing morbidity or shortening of life resulting from its excessive interactions. Chinese yam, a traditional Chinese medicinal and edible plant, has immunoregulatory functions. SCYP, a complex polysaccharide obtained from Chinese yam, was used to explore its effect on inflammation. In the present study, the administration of SCYP was found to alleviate inflammation by changing the composition of the gut microbiota. In order to simulate systemic inflammation induced by infectious injury, an intraperitoneal injection of LPS was used to establish the disease model. This model can be used to identify bioactive substances that can effectively prevent and alleviate inflammation in various parts of the organism. The body and organs have their normal weight ranges, and abnormal changes are associated with abnormal body conditions. Lipopolysaccharides can cause abnormal weight loss, possibly due to increased leptin levels [27]. Previous studies have shown that LPS causes the atrophy of the thymus and the enlargement of the spleen [26,28]. Splenomegaly is a common symptom of immune disorders and infectious inflammation, and this study found that SCYP can alleviate splenomegaly by establishing that SCYP can reduce the spleen index in cy-induced immunosuppressed mice [19]. Abnormal liver weight is a very obvious sign of the development of liver disease, and LPS not only causes the enlargement of the liver, but also its atrophy [29]. In the present study, liver weight was reduced after LPS damage, but was effectively prevented by both CYP and SCYP (Table 1).

In a study by Guo et al., LPS stimulation for 4 h was found to induce 5- to 10-fold increases in TNF-α and IL-6 in the blood, numbers that were even higher in this study after stimulation for 12 h [30]. Increased inflammatory cytokines in the blood, especially TNF-α and IL-6, are the main features of LPS-induced acute inflammation in mice [30]. TNF-α is a multifunctional cytokine that increases vascular permeability, induces fever, has potent pro-inflammatory effects, and plays a role in the induction of other inflammatory factors [31,32]. TNF-α is positively associated with a variety of inflammatory diseases, such as rheumatoid arthritis, acute liver injury, and lung cancer [30]. IL-6 is a more complex cytokine because it can be produced by immune and non-immune cells in multiple organ systems and act on multiple organ systems [31,33]. IL-6 has proinflammatory and pyrogenic functions, and overproduction exacerbates local or systemic inflammation [30]. IL-6 and TNF-α are always present together, showing the same trends, and are likewise positively correlated with various inflammatory diseases. SCYP can effectively reduce the concentration of TNF-α and IL-6 in the blood, indicating that both polysaccharides can inhibit the secretion of systemic pro-inflammatory factors and exert an anti-inflammatory effect (Figure 2).

Intraperitoneal injection of LPS can cause very typical acute liver injury and inflammation in mice. LPS, a typical pathogen-associated molecular pattern (PAMP), binds to pattern-recognition receptors (PRRs) on hepatic immune cells to initiate inflammatory responses after transport to the liver via portal blood [34]. In the response process, the lipopolysaccharide stimulates the immune cells to release a variety of inflammatory factors and chemokines which accelerate the recruitment of neutrophils, macrophages, and other immune cells, resulting in the release of more pro-inflammatory factors, including TNF-α, IL-6, and IL-1β [35,36,37]. Variations in IL-1β concentrations are associated with pathophysiological changes in different disease states and can be used to monitor disease progression [36,38]. In the present study, stimulation of LPS greatly increased the release of pro-inflammatory cytokines from the liver. Although CYP did not decrease the secretion of TNF-α in the liver, it successfully reduced the production of IL-6 and IL-1β. SCYP effectively reduced the production of three pro-inflammatory factors and was more effective than CYP. One study showed that 50 mg/kg sulfated *Cyclocarya paliurus* polysaccharides reduced IL-6 to normal levels in the liver, and in this study, 100 mg/kg SCYP also reduced IL-6 to normal levels (12 ng/mL). Sulfated polysaccharides were more effective than natural polysaccharides in both studies [29]. Overall, both CYP and SCYP could alleviate the inflammatory response of the liver caused by LPS, and the effectiveness of SCYP is superior (Figure 3).

The α-diversity analysis revealed that the LPS treatment did not cause a difference between the mice in the LPS group and those in the normal group (Figure 4), a result which is consistent with those of other studies [39]. CYP increased three indices, indicating its ability to increase the diversity of the microbial community, and SCYP did not show this effect. In the β-diversity analysis, it was seen that treatment with both yam polysaccharides could partially change the community structure, distinguishing it from the model group (Figure 4).

At the phylum level, *Bacteroidetes* and *Firmicutes* are the two most abundant bacteria in the normal intestine [40], and the ratio of *Bacteroidetes* to *Firmicutes* increased after LPS treatment, indicating that the homeostasis of the organism’s intestinal environment was disrupted, resulting in an imbalance [41]. The addition of polysaccharides slightly reversed this imbalance. The abundance of *Proteobacteria* increased after LPS treatment. Most *Proteobacteria* are pathogenic gram-negative bacteria, and as markers of dysbiosis in the intestinal flora, their abnormally elevated levels are associated with inflammatory diseases [39]. There was a tendency for *Proteobacteria* to decrease after treatment with yam polysaccharides (Figure 5). Sulfated *Cyclocarya paliurus* polysaccharides reduced the relative abundance of *Proteobacteria* to 5% in one study, and SCYP was similarly reduced to 5% in this study [40].

At the genus level, two yam polysaccharides have the effect of altering the relative abundance of certain genera. *Shigella* is a kind of pathogenic bacteria belonging to the *Enterobacteriaceae* family which is positively correlated with metabolic endotoxin levels and the severity of systemic inflammation, and its increase may lead to intestinal barrier damage and increased permeability, exacerbating bacterial invasion and translocation [42,43]. *Desulfovibrio*, a kind of gram-negative conditional pathogenic bacteria, is classified as a genus of pro-inflammatory bacteria because of its high endotoxin content [44]. *Desulfovibrio* can reduce sulfate to produce H_2_S, which can trigger epithelial apoptosis and intestinal barrier damage, and is associated with obesity and inflammation [45,46]. *Sutterella* is an important kind of commensal bacteria found in the intestinal tract, and changes in its levels have been associated with diseases including autism, depression, Down syndrome, obesity, and inflammatory bowel disease [47,48,49,50]. *Sutterella*, a member of the *Proteobacteria* classification, is abnormally increased in intestinal flora disorders [49]. Numerous studies have shown that intraperitoneal injection of LPS causes an increase in the genera *Shigella*, *Desulfovibrio*, and *Sutterella*, and the results of the present study are consistent with this observation [51]. Pretreatment with both polysaccharides had a tendency to reduce *Shigella*. Both CYP and SCYP were effective in reducing *Desulfovibrio* to normal levels, and SCYP showed better effects. Additionally, CYP effectively inhibited *Sutterella*. The above results indicate that both polysaccharides inhibit the growth of harmful bacteria with different strengths.

*Prevotella* is a dominant genus of bacteria in the human intestine which is associated with plant-based diets rich in carbohydrates and fiber, and which helps to break down polysaccharides [52,53]. *Prevotella* can produce propionic acid, which has the effect of preventing obesity and reducing the risk of diabetes [53,54]. CYP increased *Prevotella*, indicating that CYP effectively changed the mouse enterotype so that the mouse intestine could better utilize polysaccharides to produce SCFAs. *Coprococcus* can synthesize acetate and butyrate. Acetate may be a source of energy for endurance exercise, and butyrate is not only the main source of energy for colonic cells but can also play an anti-inflammatory role [20,55,56,57]. The experimental results suggest that SCYP may have a beneficial effect on intestinal health by increasing the proportion of *Coprococcus* in the intestine. These results suggest that the two polysaccharides regulate the intestinal flora by promoting the growth of different beneficial bacteria.

The LEfSe analysis revealed that, in the LPS group, *Proteobacteria*, *Gammaproteobacteria*, *Enterobacterianceae*, *Enterobacteriales*, and *Shigella* were the dominant species, and most of these are pathogenic or conditionally pathogenic bacteria [39,42]. *Bacteroidia*, *Bacteroidales*, *Prevotellaceae*, *Prevotella, Flavobacteriaceas*, *Flavobacteriales*, and *Flavobacteriia* were biomarkers of CYP and are associated with the degradation of biopolymers [58,59]. This indicates that CYP promoted the proliferation of polysaccharide-degrading bacteria and facilitated the intestinal de-utilization of polysaccharides. *Bacteroidaceae*, *Bacteroides*, and *Pasteurella* became SCYP biomarkers. *Bacteroidaceae* was the dominant bacterial family, indicating that SCYP promoted the proliferation of polysaccharide-utilizing bacteria. *Pasteurella* appeared somewhat anomalously, as was the case in a cactus polysaccharide study in which *Pasteurella* became the dominant bacterium in the polysaccharide group [60].

Substrate specificity is a key facet of the microbial response to complex carbohydrates. Thus, polysaccharides with different structures will be fermented with different kinds of microbiota, resulting in a new component of the gut microbiota. The present study revealed that the two polysaccharides selectively modulated the composition of the gut microbiota, among which SCYP produced different regulatory effects on the gut microbiota and exerted different anti-inflammatory effects from those of CYP, possibly because of its higher number of sulfate groups, higher galactose content, and larger Mw [25]. The effect of the intestinal flora was mediated via the gut–liver axis, where SCYP had a better prophylactic and anti-inflammatory effect than CYP, probably because SCYP was better able to indirectly act on the hepatic receptors to produce a response. The intestinal flora results showed that, overall, CYP was more able than SCYP to promote beneficial bacteria to become the dominant bacteria, a result consistent with the better effect of CYP on alleviating inflammation in the blood circulation. Our previous study showed that sulfation modification could enhance the immunomodulatory activity of yam polysaccharides [19,25], and this present study has shown that sulfation modification can improve the anti-inflammatory activity of yam polysaccharides in certain organs, supporting the notion that chemical modification has different effects on different biological activities. To clarify the differential effects of CYP and SCYP, further research on the anti-inflammatory molecular mechanisms of both should be studied.

## 5. Conclusions

The present study has demonstrated that sulfated yam polysaccharide has preventive alleviating effects on LPS-induced systemic acute inflammation, including the restoration of body weight and organ index and the reduction of inflammatory factors (TNF-α, IL-6, and IL-1β) released in the blood and liver, which may be associated with maintaining normal intestinal flora and reversing intestinal flora imbalance. CYP and SCYP reduced pathogenic bacteria; CYP increased *Prevotella*, and SCYP increased *Coprococcus*. CYP and SCYP have different effects on the regulation of the intestinal flora and have different advantageous anti-inflammatory effects on different organ systems, which can be attributed to the changes in molecular structure after sulfated modification. This study also showed that both polysaccharides, as natural and modified active substances, can be expected to improve the health of the body when used as prebiotics, and that they may also have the potential to be used as prophylactic agents for the treatment of inflammatory diseases. The molecular mechanisms involved in the sulfated yam polysaccharides can be further researched.

## Figures and Tables

**Figure 1 foods-12-01772-f001:**
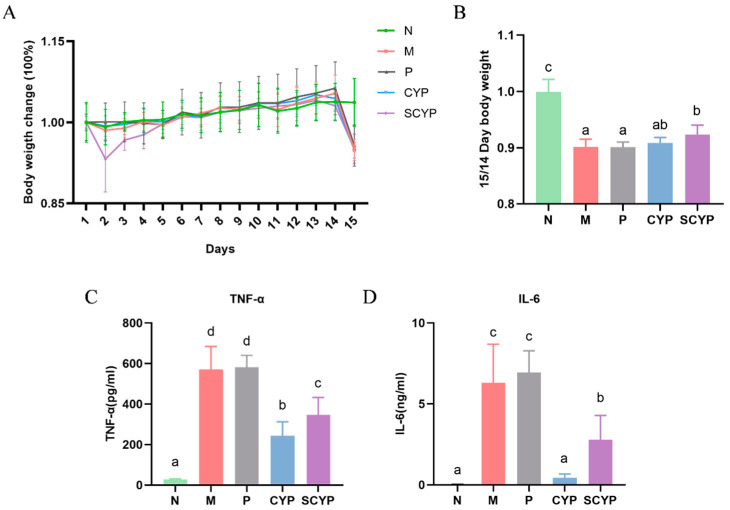
Percentage change in body weight compared with day 1 for each group (**A**). Body weight ratio of day 14/15 for each group (**B**). Levels of pro-inflammatory cytokines TNF-α (**C**) and IL-6 (**D**) in the plasma. Values are expressed as means ± SD. (n = 6 for **A**,**B** and n = 5 for **C**,**D**). Different letters represent significant differences between different groups (*p* < 0.05).

**Figure 2 foods-12-01772-f002:**
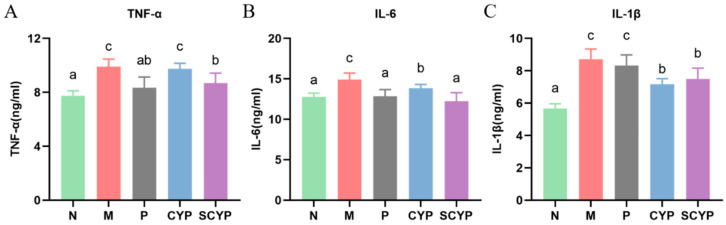
Influence of CYP and SCYP on pro-inflammatory cytokines TNF-α (**A**), IL-6 (**B**) and IL-1β (**C**) in the livers of LPS-induced mice. Values are expressed as means ± SD. (n = 6). Different letters represent significant differences between different groups (*p* < 0.05).

**Figure 3 foods-12-01772-f003:**
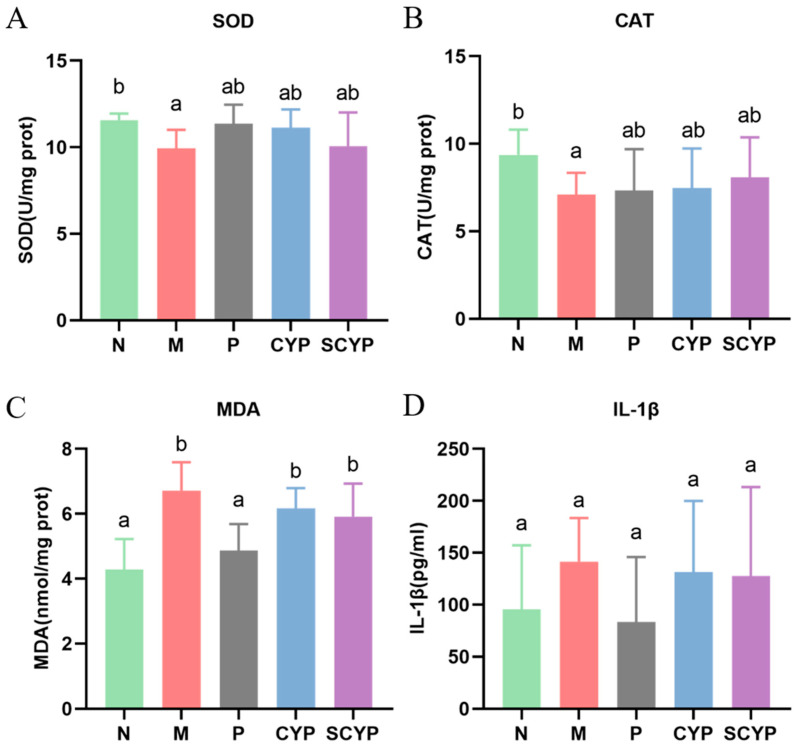
Oxidative stress in jejunal tissue evaluated in terms of SOD (**A**), CAT (**B**), and MDA (**C**). Levels of pro-inflammatory cytokines IL-1β (**D**) in jejunal tissue. Values are expressed as means ± SD. (n = 6 for **A** and n = 5 for **B**,**C**). Different letters represent significant differences between different groups (*p* < 0.05).

**Figure 4 foods-12-01772-f004:**
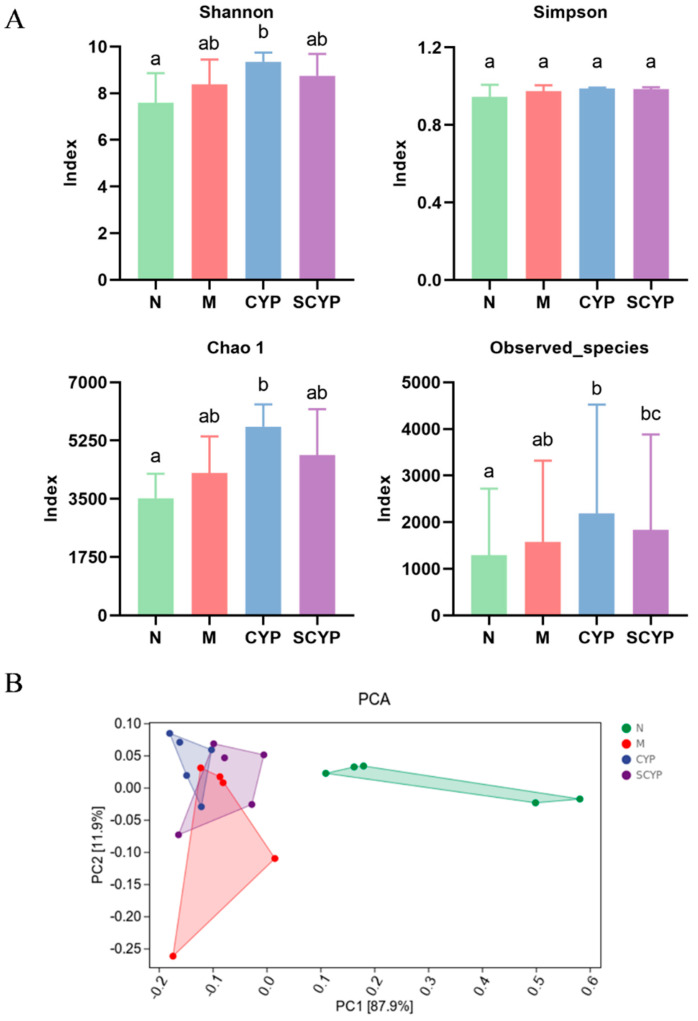
α-diversity evaluated using Shannon, Simpson, Chao1, and observed_species indices (**A**). β-diversity evaluated according to the PCA (**B**). (n = 5). Different letters represent significant differences between different groups (*p* < 0.05).

**Figure 5 foods-12-01772-f005:**
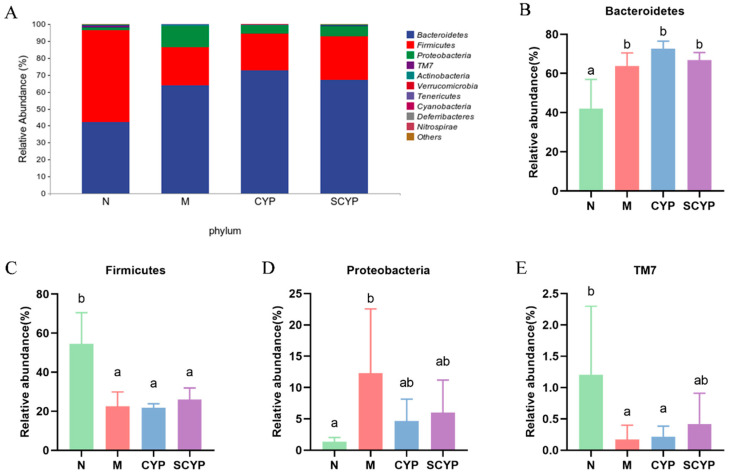
Species composition of colon microbiota at the phylum level (**A**). Relative abundance of colon microbiota at the phylum level (**B**–**E**). Values are expressed as means ± SD. (n = 5). Different letters represent significant differences between different groups (*p* < 0.05).

**Figure 6 foods-12-01772-f006:**
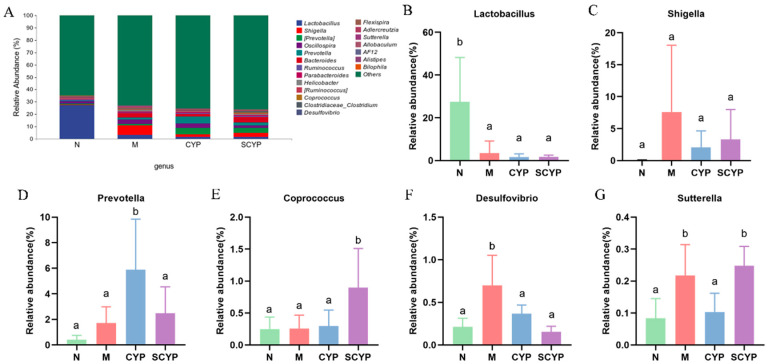
Species composition of colon microbiota at the genus level (**A**). Relative abundance of colon microbiota at the genus level (**B**–**G**). Values are expressed as means ± SD. (n = 5). Different letters represent significant differences between different groups (*p* < 0.05).

**Figure 7 foods-12-01772-f007:**
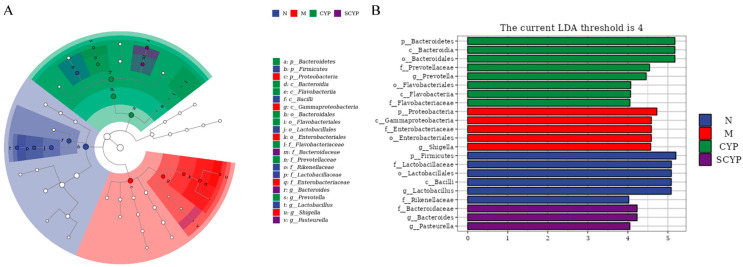
Taxonomic cladogram of the LEfSe analysis (**A**). Distribution histogram of the LEfSe analysis (**B**). The LDA threshold is 4. (n = 5).

**Table 1 foods-12-01772-t001:** The organ indices of the LPS-induced mice.

	Spleen Index	Thymus Index	Liver Index
N	2.64 ± 0.16 a	1.57 ± 0.07 a	30.42 ± 0.89 b
M	4.02 ± 0.23 c	1.33 ± 0.19 a	28.09 ± 1.07 a
P	4.06 ± 0.30 c	1.33 ± 0.24 a	31.19 ± 0.46 b
CYP	3.92 ± 0.10 c	1.36 ± 0.26 a	30.23 ± 0.60 b
SCYP	3.61 ± 0.27 b	1.35 ± 0.08 a	30.33 ± 2.27 b

Values are expressed as means ± standard deviation (SD). Different lowercase letters represent significant differences between different groups (*p* < 0.05) (n = 5).

## Data Availability

The datasets generated for this study are available on request to the corresponding author.

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
