# Peer review of "Sulfated Chinese Yam Polysaccharides Alleviate LPS-Induced Acute Inflammation in Mice through Modulating Intestinal Microbiota"

_foods, 2023, doi:10.3390/foods12091772_

Round 1

Reviewer 1 Report

Reviewer comments and suggestions 

The authors in this study investigated the preventive anti-inflammatory properties of Chinese yam polysaccharides (CYP) and sulfated Chinese yam polysaccharides (SCYP) on LPS-induced systemic acute inflammation in mice. The authors reported that SCYP can competently reduce plasma TNF-α and IL-6 levels, which exhibited anti-inflammation ability better than CYP. Linear discriminant analysis (LDA) effect size (LEfSe) showed that the treatment with CYP and SCYP could produce more biomarkers of the gut microbial that can promote the proliferation of polysaccharide-degrading bacteria and facilitate the intestinal de-utilization of polysaccharides.

Minor Comments

  1. In Typo error in line 19, please check it
  2. Line 39-40 The line needs more references.
  3. Line 58-59 What do the authors want to say here?
  4. Line 102 Please rewrite the sentence.
  5. Line 145 -147 The legend should be indicated correctly.
  6. Line 151-152 what was a, b, c indicated in Table 1, please mention it in the legend part)
  7. Line 183 The authors need to add a basic para for this analysis for the common reader of your manuscript.
  8. Line 241-243 The first para should be the novelty of your study not repeating the same line or citing other references
  9. Line 252-253 Move these lines upward (first paragraph of discussion).
  10.  Line 274-278 The authors should indicate tables or figures after discussing the concerned results.
  11. Line 318 It would be nice if the authors could explain the growing condition differences or any specific reason for both for showing the different properties. Polysaccharide composition is also required for the selective findings
  12. All references should be modified based on MDPI journals. 

Reviewer 2 Report

This work is devoted to the study of Chinese yam sulfated polysaccharides and their anti-inflammatory effect. The article is written clearly and accessible, the main ideas and experimental data are beyond doubt. The topic of this study is relevant, since polysaccharides and their derivatives, as well as their biologically active properties, have been actively studied recently. Natural polysaccharides and their derivatives have a number of significant advantages over their synthetic counterparts. Despite the relevance and other advantages of this study, there are some points for improvement:

1. Sulfated polysaccharides have many useful biological properties: anticoagulant, hypolipidemic and others. This must be written in the introduction.

2. It is necessary to clarify in the text what kind of polysaccharides are included in the composition of the yam, which the authors are studying. In addition, it is necessary to clarify the composition of monosaccharides in these polyaccharides.

3. A more detailed comparison of the obtained results with data from the literature is required.

4. Please cite: 10.1007/s13399-021-02250-x.

5. It is desirable to expand the conclusions.
